# The Role of the Interface of PLA with Thermoplastic Starch in the Nonisothermal Crystallization Behavior of PLA in PLA/Thermoplastic Starch/SiO_2_ Composites

**DOI:** 10.3390/polym15061579

**Published:** 2023-03-22

**Authors:** Deling Li, Congcong Luo, Jun Zhou, Liming Dong, Ying Chen, Guangtian Liu, Shuyun Qiao

**Affiliations:** 1College of Materials and Chemical Engineering, Xuzhou University of Technology, Xuzhou 221018, China; 2School of Environment and Chemical Engineering, Yanshan University, Qinhuangdao 066004, China; 3College of Electrical and Control Engineering, Xuzhou University of Technology, Xuzhou 221018, China

**Keywords:** polylactide, composites, interfacial binding energy, nonisothermal crystallization kinetics

## Abstract

Corn starch was plasticized by glycerol suspension in a twin-screw extruder, in which the glycerol suspension was the pre-dispersion mixture of glycerol with nano-SiO_2_. Polylactide (PLA)/thermoplastic starch/SiO_2_ composites were obtained through melt-blending of PLA with thermoplastic starch/SiO_2_ in a twin-screw extruder. The nonisothermal crystallization behavior of PLA in the composites was investigated by differential scanning calorimetry. An interface of PLA with thermoplastic starch was proven to exist in the composites, and its interfacial bonding characteristics were analyzed. The interfacial binding energy stemming from PLA with thermoplastic starch exerts a significant influence on the segmental mobility of PLA at the interface. The segmental mobility of PLA is gradually improved by increasing interfacial binding energy, and consequently, the relative crystallinity on the interface exhibits progressive promotion. The Jeziorny model could well describe the primary crystallization of PLA in the composites. The extracted Avrami exponents based on the Jeziorny model indicate that the primary crystallization of PLA follows heterogeneous nucleation and three-dimensional growth. This study has revealed the intrinsic effect of the interfacial segmental mobility on the nonisothermal crystallization behavior of PLA in composites, which is of technological significance for its blow molding.

## 1. Introduction

PLA is an eco-friendly material due to its renewability and biodegradability. Disposable film prepared by PLA is suitable as a sustainable alternative to petrochemical-based film, which is beneficial for the alleviation of the consumption of oil, as well as the eradication of the environmental pollution induced by the petrochemical-based film [1,2,3,4,5]. However, the production costs of disposable PLA film are much higher than those of polyethylene and polypropylene, which limits its extensive application. Except for the cost of PLA, the low melt strength and poor viscoelastic behavior of PLA due to its high critical entanglement molecular weight also seriously limit its processing using extrusion blow [6,7,8,9,10].

To produce PLA-based composites with a lower cost and high melt strength, PLA/starch/SiO_2_ composites were prepared in our previous work [11,12]. Starch is commonly considered as an available bio-based material due to its lower cost and its biodegradability and renewability. The incorporation of PLA with starch could distinctly reduce the cost of disposable PLA film [11,12,13,14,15]. The microstructure of a sandbag of nano-SiO_2_ encapsulated by thermoplastic starch and the interfacial characteristics of PLA with thermoplastic starch could account for its enhanced rheological properties and tensile strength [11]. However, it still remains a specific challenge to evaluate the nonisothermal crystallization behavior of PLA/starch/SiO_2_ composites. Generally, the processing of polymers is commonly carried out under nonisothermal conditions. For crystalline polymers, the nonisothermal crystallization behavior of polymers has a crucial effect on their tensile strength and microstructures [16,17,18,19,20]. Fine crystalline grains formed during melt blow molding could act as chain junctions to increase their melt strength, and consequently, their enhanced melt strength could maintain the stability of film bubble. In addition, the tensile strength could be improved by crystalline grains of appropriate size. Thus, it is necessary to obtain a fundamental understanding of how the interfacial characteristics of PLA/thermoplastic starch dominate the crystallization behavior.

The crystallization kinetics of polymers is a widely accepted means to evaluate their crystallization behavior. Many mathematical models are proposed to clarify the macroscopic evolution of crystallinity for polymers during their crystallization process [21,22,23,24,25]. One of the most frequently used models to describe crystallization kinetics is the Avrami equation with the assumptions of nucleation and geometric growth occurring during the crystallization process of the polymers [26,27,28,29]. For the analysis of nonisothermal crystallization under linear heating conditions, Jeziorny developed the conventional Avrami equation by adjusting the linear Avrami equation to linear heating conditions by introducing a constant heating rate into the rate constant [30,31,32]. The Jeziorny model is a widely used method to study nonisothermal crystallization and is well suitable for the analysis of the corresponding primary crystallization [17,25,33,34,35,36].

In this study, the nonisothermal crystallization behavior of PLA in the composites was investigated by differential scanning calorimetry (DSC). The interfacial bonding energy of PLA with thermoplastic starch was calculated by Materials Studio. The effect of the interface of PLA with thermoplastic starch on the nonisothermal crystallization behavior of PLA in the composites is discussed.

## 2. Materials and Methods

### 2.1. Materials

Nano-SiO_2_ with a specific surface area of 200 ± 25 m^2^·g^−1^ was purchased from Shanghai Yizhu Industrial Co., Ltd. (Shanghai, China). PLA with $\overset{-}{M_{W}}$ of 8.0 × 10^4^ g/mol and polydispersity of 2.1 was purchased from Shanghai Macklin Biochemical Co., Ltd. (Shanghai, China). Corn starch of reagent grade was purchased from Shanghai Aladdin Bio-Chem Technology Co., Ltd. (Shanghai, China). Nano-SiO_2_, PLA, and corn starch were dried under a vacuum at 50 °C for 12 h before use. Glycerol of analytical grade was purchased from Tianjin Fuyu Chemical Co., Ltd. (Tianjin, China) and used as received.

### 2.2. Preparation of Thermoplastic Starch/SiO_2_

The process was conducted as follows: 40 parts of glycerol and 2 parts of nano-SiO_2_ were first mixed and sonicated for 15 min and 100 parts of corn starch were mixed into the suspension of nano-SiO_2_ with glycerol and strongly stirred for 1 h at a speed of 2000 rpm. The obtained mixtures were kept for 3 h at room temperature, and subsequently, the thermoplastic starch/SiO_2_ composites were prepared through melt plasticization using a twin-screw extruder (czs-shj-20, Nanjing Yongjie Chemical Machinery Co., Ltd. (Nanjing, China)). The temperature profile from zone 1 to zone 6 was as follows: 115 °C, 120 °C, 130 °C, 125 °C, 120 °C, and 120 °C in the extruder barrel, and 120 °C in the extruder head die. The screw speed was 150 rpm. The obtained thermoplastic starch/SiO_2_ composites were granules.

The same procedure was performed except that the mass fraction of nano-SiO_2_ was 4, 6, and 8 parts, respectively.

### 2.3. Preparation of PLA/Thermoplastic Starch/SiO_2_ Composites

One hundred parts of PLA and fifteen parts of thermoplastic starch/SiO_2_ were mixed and then extruded using a twin-screw extruder. The mass fraction of nano-SiO_2_ in thermoplastic starch/SiO_2_ varied from 2 to 8 parts. The temperature profile from zone 1 to zone 6 was as follows: 155 °C, 165 °C, 160 °C, 150 °C, 145 °C, and 140 °C in the extruder barrel, and 140 °C in the extruder head die. The screw speed was 150 rpm. The obtained PLA/thermoplastic starch/SiO_2_ composites were granules.

The same procedure was performed except that the mass fraction of thermoplastic starch/SiO_2_ was 10, 15, 20, 25, and 30 parts, respectively. Here, the mass fraction of nano-SiO_2_ in thermoplastic starch/SiO_2_ was six parts.

### 2.4. Characterizations

All of the tested samples were dried under a vacuum at 50 °C for 12 h before the test.

All of the DSC measurements were conducted on a Perkin-Elmer DSC system under N_2_ atmosphere and involved two heating processes. About 6 mg of the sample was first heated from −50 to 200 °C at a heating rate of 10 °C·min^−1^ and kept at 200 °C for 3 min, then cooled to −50 °C at a cooling rate of −100 °C·min^−1^ and kept at −50 °C for 20 min. The second heating was from −50 to 200 °C at a heating rate of 10 °C·min^−1^. The information on the thermal history obtained from the first heating process mainly involved thermoplastic processing and was eliminated after thermostatic control at 200 °C. There was no obvious exothermic peak observed during the cooling process at the cooling rate of 100 °C·min^−1^, which suggested that the tested sample could not crystallize under this cooling condition. The rapid cooling process allowed us to analyze the thermal behavior of the tested sample from the amorphous state during the second heating process. The second heating process could provide information about the behavior of the chain segment. The nonisothermal crystallization kinetics were analyzed according to DSC curves obtained from the second heating process.

The crystallinity of the composites was calculated according to Equation (1).
(1)$$X_{c}=\frac{∆H}{∆H^{*}\cdot w_{PLA}}$$
where $w_{PLA}$ is the mass fraction of PLA in the composites,∆H is the melting enthalpy of the melting endotherms obtained from the DSC curves, and $∆H^{*}$ is the melting enthalpy of 100% crystalline PLA and its value was 93 J·g^−1^ [17,37].

The relative crystallinity $X\left(t\right)$ at an arbitrary time involved in the heating crystallization process was calculated by Equation (2).
(2)$$X\left(t\right)=\frac{Q_{t}}{Q_{t\infty }}=\frac{\int_{t_{0}}^{t}\frac{dH}{dt}dt}{\int_{t_{0}}^{t_{\infty }}\frac{dH}{dt}dt}$$
where $t_{0}$, t, and $t_{\infty }$ represent the onset, an arbitrary time, and the overall time when the crystallization was finished, respectively, $Q_{t}$ is the crystallization enthalpy released during an infinitesimal temperature range dT, and $Q_{t\infty }$ is the overall crystallization enthalpy for a specific heating condition.

### 2.5. Simulation

The molecular dynamics simulation of the interface of PLA with thermoplastic starch was conducted by Materials Studio 2020. The thermoplastic starch model and the PLA model were established with MS visualizer. The interfacial model of PLA with thermoplastic starch was constructed by an amorphous cell according to the mass fraction of PLA, AM, and Gly. The geometric optimization and calculation of these models, as well as the results analysis and output, were carried out by Forcite. In the molecular dynamics simulations and the model optimization, the compass II force field was used for all force fields. The molecular dynamics simulation of the geometrically optimized interface model was carried out at 298 K and 1 fs of the time step in the NVT ensemble. The simulation involved two stages of equilibrium and data collection. In the equilibrium stage, the initial velocity met the Maxwell–Boltzmann distribution, and an Andersen thermostat was used for the control of temperature. Subsequently, to achieve complete equilibrium, the cumulative calculation was 500 ps. In the data collection stage, another 500 PS was calculated on the basis of the last frame configuration of the equilibrium stage, in which a Nose thermostat was used, and a frame configuration was output every 5 PS. A total of 101 frame configurations (including the frame at time 0) were collected for the data analysis.

## 3. Results

### 3.1. The Elucidation of the Interface of PLA with Thermoplastic Starch Existing in the Composites

Amorphous thermoplastic starch could be obtained through the incorporation of native starch with polyol plasticizers under a thermomechanical process, in which the hydrogen bonds in native starch are partially substituted by the hydrogen bonds of polyol [13,14,15]. In this research, to achieve the uniform dispersion of nano-SiO_2_ in starch, nano-SiO_2_ was dispersed in glycerol by ultrasonic dispersion. Then, the plasticization process of starch with glycerol was conducted by melt extrusion. Subsequently, PLA/thermoplastic starch/SiO_2_ composites were obtained by melt extrusion of PLA with thermoplastic starch/SiO_2_.

To evaluate the effect of SiO_2_ on the segmental mobility of PLA, the mass fraction of PLA and thermoplastic starch/SiO_2_ is in 100 parts and 15 parts, respectively, while the mass fraction of SiO_2_ in thermoplastic starch/SiO_2_ varies from 2 to 8 parts. DSC curves of these composites obtained from the second heating process are shown in Figure 1. For thermoplastic starch available via melt extrusion, it has been reported that its crystalline structure is destroyed and it exhibits an amorphous state [14,15]. The DSC curves in Figure 1 have no obvious heat flow signal of crystallization or melting t thermoplastic starch, which suggests that the crystallization and melting peaks are attributed to PLA. For T_g_ of neat thermoplastic starch plasticized by glycerol, its value is about 28 °C [13,16]. There is only one T_g_ in the DSC curve, which indirectly indicates that PLA has good compatibility with thermoplastic starch. The thermal transition temperatures mainly reflect the segmental mobility of PLA. When the mass fraction of PLA and thermoplastic starch are the same in the composites, the T_g_, T_c_, T_m1_, and T_m2_ of the composites do not change significantly with the mass fraction of SiO_2_ in thermoplastic starch. It is inferred that SiO_2_ is mainly wrapped in starch, and thus fails to form an effective interface of PLA/SiO_2_. SiO_2_ wrapped in starch has no obvious effect on the segmental mobility of PLA.

Quite contrary to SiO_2_, an interface of PLA with thermoplastic starch could probably be formed in the composites. The T_g_, T_c_, T_m1_, and T_m2_ of the composites sharply decreased as compared with those of neat PLA, suggesting that thermoplastic starch might exert an influence on the segmental mobility of PLA. When the mass fraction of PLA and thermoplastic starch are the same, the composites exhibit similar thermal transition temperatures and crystallization temperatures due to their similar interface of PLA with thermoplastic starch. This is consistent with the rheology and mechanical properties of the composites reported earlier. Nano-SiO_2_, mainly wrapped by thermoplastic starch, forms a “sandbag” microstructure [11]. It is dispersed in the PLA matrix, which prevents the formation of cracks and continuous development in the matrix and thus effectively improves the mechanical strength of the composites.

### 3.2. The Effect of the Interface of PLA with Thermoplastic Starch on the Thermal Transition Temperature of the Composites

To evaluate the effect of the interface of PLA with thermoplastic starch on the crystallization of PLA in the composites, the mass fraction of components of the composites is 100 parts of PLA, and for thermoplastic starch/SiO_2_, it is 10, 15, 20, 25 and 30 parts, respectively. In this part, the thermoplastic starch comprises 100 parts of starch, 40 parts of glycerol, and 6 parts of SiO_2_.

The interchain free volume of PLA is decreased due to the pressure in the barrel during the melt extrusion process. The segmental mobility of PLA needs a higher temperature, which leads to a higher T_g_ for the composite with a thermal history. Thus, for the same mass fraction of thermoplastic starch in the composite, the T_g_ of the composites with a thermal history is higher than that without a thermal history, as shown in Table 1. Besides the effect of the pressure on T_g_, the hydrogen bonds between thermoplastic starch and PLA also exert an effect on the segmental mobility of PLA. The T_g_ of the composite with a thermal history increased slightly with the increasing mass fraction of thermoplastic starch due to its increasing number of hydrogen bonds, and for the composites without a thermal history, the T_g_ varied obviously. It is well accepted that interchain hydrogen bonds could improve the T_g_ of the polymer [17,38].

As shown in Figure 2, the T_g_ of the composite is much lower than that of neat PLA because of the plasticizing effect of thermoplastic starch on PLA. Compared with neat PLA, the T_m_ of the composites decreased significantly. The main melting peak (T_m2_) is accompanied by a shoulder peak (T_m1_), suggesting that thermoplastic starch could act as the crystallization nucleator during the heating process of the composites and stimulate PLA to crystallize on the interface of PLA with thermoplastic starch at a lower temperature. The shoulder peak is ascribed to the contribution of the melting of the interfacial crystal, and the main peak is ascribed to the contribution of the melting of the crystal in the PLA matrix. T_m1_ increases with the increasing mass fraction of thermoplastic starch due to the increasing interfacial area. Unlike T_m1_, T_m2_ undergoes almost no change with the mass fraction of thermoplastic starch, suggesting that the crystal in the matrix exhibits similar crystalline perfection.

### 3.3. The Crystallizability of PLA in the Composites

The considered composites are also composed of 100 parts of PLA and variable thermoplastic starch/SiO_2_ with 10, 15, 20 25, and 30 parts, in which the thermoplastic starch is 100 parts of starch, 40 parts of glycerol, and 6 parts of SiO_2_. Compared with neat PLA, the crystallization of PLA in the composites during its second heating process happens at a lower temperature. Just as shown in Table 2, the T_c,onset_ and T_c_ of the composites are obviously lower than that of the PLA. At the same time, the crystallinity is significantly improved with the increasing mass fraction of thermoplastic starch. This suggests that the heterogeneous nucleation exerted by thermoplastic starch has an appreciable effect on the crystallization of PLA in the composites.

PLA would crystallize at its interface with thermoplastic starch and in its matrix, in which heterogeneous nucleation and growth of PLA occur on the interface at a lower temperature. This crystallization behavior could be furtherly clarified by the main melting peak and the shoulder peak derived from the melting endotherms on the DSC curves. The curve-fitting of the melting endotherms is shown in Figure 3. The main melting peak is assessed as the melting peak of the PLA crystal in the matrix, and the shoulder peak is related to the PLA crystal on the interface. $∆H_{p,m}$ and $∆H_{p,s}$ are the melting enthalpy of the main melting peak and the shoulder peak, respectively. The relative proportion of crystallization on the interface ($X_{c,interf}$) or in the PLA matrix ($X_{c,m}$) is calculated according to the following Equations (3) and (4), respectively.
(3)$$X_{c,interf}=\frac{∆H_{p,s}}{∆H}$$
(4)$$X_{c,m}=\frac{∆H_{p,m}}{∆H}$$

With the increasing mass fraction of thermoplastic starch, $X_{c,interf}$ increases, but $X_{c,m}$ decreases. This illustrates that the interfacial area of PLA with thermoplastic starch gradually increases due to the increasing mass fraction of thermoplastic starch. Consequently, at their interface, the number of hydrogen bonds between PLA and thermoplastic starch chains increased, as well as their interchain forces.

### 3.4. Nonisothermal Crystallization Kinetics Analysis

The Jeziorny model extended directly the Avrami equation from the isothermal crystallization to the nonisothermal crystallization in the following form:(5)$$log\left[-ln\left(1-X\left(t\right)\right)\right]=logZ+nlogt$$
where $X\left(t\right)$ is the relative crystallinity, n is the Avrami exponent related to the growth geometry of the crystals, and Z is the crystallization rate constant depending on nucleation and growth rate.

Considering the nonisothermal character of the crystallization in a constant heating rate, the crystallization rate constant is adjusted by introducing a constant heating rate, and the corrected crystallization rate constant ($Z_{c}$) is shown as follows:(6)$$logZ_{c}=\frac{logZ}{\phi }$$
where ϕ is the heating rate.

Generally, the Jeziorny model cannot simultaneously describe the primary crystallization and the secondary crystallization of the composites well. Still, it has a practical implication for the primary crystallization behavior at a nonisothermal crystallization condition [17,25,33].

Here, the Jeziorny model is adopted to evaluate the primary crystallization behavior of the PLA in the composites. Figure 4 is the plots of $log\left[-ln\left(1-X\left(t\right)\right)\right]$ versus logt, as well as the fitted lines of the primary crystallization period. The curves in Figure 4 contain the linear parts from the beginning to about 50–60% of the crystallization range and the non-linear part of the deviation from the linear relation. These two parts are ascribed to the primary crystallization period and the secondary crystallization period, respectively. Obviously, there is a satisfactory fitting for every curve because of the linear correlation coefficient ($R^{2}>0.99$) indicating the linearity of the main crystallization. The crystallization rate constant and the Avrami exponent extracted from the fitted lines are listed in Table 3.

It is accepted that the Avrami exponent of 3 suggests heterogeneous nucleation and three-dimensional growth of the crystals, and the value of 4 indicates homogeneous nucleation and three-dimensional growth of the crystals [39]. The Avrami exponents of these composites are around 3.5, suggesting that the nonisothermal crystallization of PLA on the interface might involve heterogeneous nucleation and three-dimensional growth. The primary crystallization of the PLA chain occurs in the PLA matrix and at the interface of PLA/thermoplastic starch, as discussed in the part on the crystallizability of the PLA in the composites. The thermoplastic starch as a plasticizer could enhance the segmental mobility of PLA, subsequently promoting the crystallization rate due to the reduction in the energy required for the chain folding process during crystallization. This is consistent with the discussion of PLA plasticized with jojoba oil [17]. The Avrami exponents are almost unchangeable with the varied mass fraction of thermoplastic starch, which suggests that the composites have a similar nucleation mechanism. The crystallization rate constant increases slightly with the increasing mass fraction of thermoplastic starch, suggesting that the thermoplastic starch exerted some effects on the crystallization of PLA.

### 3.5. The Interfacial Binding Energy of PLA with Thermoplastic Starch

The interchain forces of PLA with thermoplastic starch on their interface are assumed to be the root reason for the nucleation mechanism of PLA in the composites. In order to investigate these interfacial interchain forces, the MD simulation is employed to construct an interfacial molecular structure model and calculate its interfacial binding energy ($E_{int}$). The interfacial binding energy of PLA with thermoplastic starch exhibits a steady increase with the increasing mass fraction of thermoplastic starch, suggesting that a composite with a higher mass fraction of thermoplastic starch would have a higher interfacial area of interaction. The increasing interfacial area of interaction would not only stabilize the crystal nucleus but also speed up the segmental mobility of the PLA. Consequently, the crystallization of PLA is promoted to a certain extent. This is well consistent with the experimental phenomenon in that the relative proportion of crystallinity on the interface increases with the increase in the fraction of thermoplastic starch.

The interfacial binding energy consists of electrostatic force and van der Waals force, in which the two forces account for almost the same contributions (~50% vs. ~50%) for each mass fraction of thermoplastic starch in the composites, as shown in Figure 5. For the electrostatic force, the hydrogen bond (H-bond) energy of PLA with thermoplastic starch has a dominant proportion. The relative H-bond proportion of PLA with AM is around 70%, and 30% for PLA with glycerol, just as shown in Figure 6. It can be inferred that the H-bond energy of PLA with AM has an essential effect on the crystallization of the PLA on the interface.

## 4. Conclusions

There is almost no interface of PLA/SiO_2_ formed in the PLA/thermoplastic starch/SiO_2_ composites. Nano-SiO_2_ wrapped by thermoplastic starch has little effect on the nonisothermal crystallization of PLA. The interface of PLA with thermoplastic starch is effective in the composites, and its interfacial area increases with the increasing mass fraction of thermoplastic starch. The interface of PLA with thermoplastic starch exerted a remarkable effect on the segmental mobility of PLA on the interface. The T_g_ and T_m_ of the composites are much lower than those of pure PLA. The relative proportion of crystallization on the interface increases with the mass fraction of thermoplastic starch. Still, the varied trend for the crystallization of PLA in the matrix is on the contrary. The Jeziorny model could describe well the primary crystallization of PLA in the composites. The extracted Avrami exponents based on the Jeziorny model indicate that the crystallization of PLA on the interface follows heterogeneous nucleation and three-dimensional growth. The relative proportion of crystallinity on the interface increases with the increasing the fraction of thermoplastic starch, which could be well explained by the interfacial binding energy of PLA with thermoplastic starch. The composite with a higher mass fraction of thermoplastic starch would have a higher interfacial area of interaction and, consequently, have a higher relative proportion of crystallinity on the interface.

## Figures and Tables

**Figure 1 polymers-15-01579-f001:**
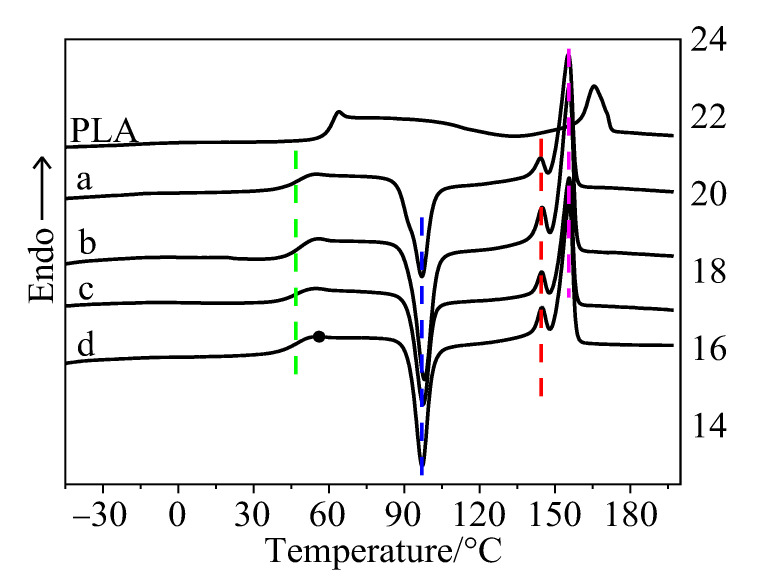
DSC curves of neat PLA and the PLA/thermoplastic starch/SiO_2_ composite without a thermal history. Curve (a) is for two parts, (b) is for four parts, (c) is for six parts, and (d) is for eight parts of SiO_2_ in thermoplastic starch.

**Figure 2 polymers-15-01579-f002:**
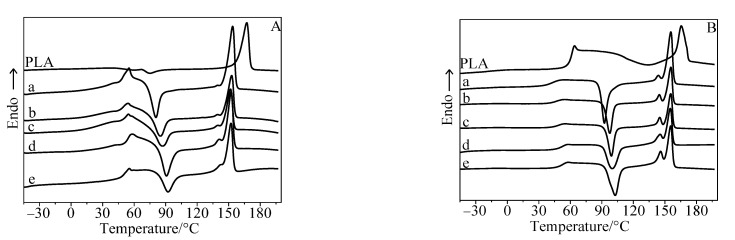
DSC curves of neat PLA and the PLA/thermoplastic starch/SiO_2_ composite. (**A**) The composites with a thermal history; (**B**) the composites without a thermal history. Curve (a) is for 10 parts, (b) is for 15 parts, (c) is for 20 parts, (d) is for 25 parts, and (e) is for 30 parts of thermoplastic starch.

**Figure 3 polymers-15-01579-f003:**
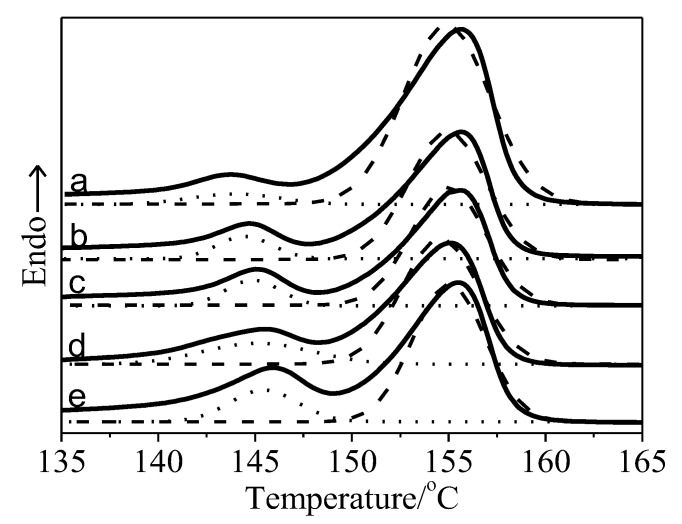
DSC curves of neat PLA and the PLA/thermoplastic starch/SiO_2_ composite. The solid lines represent the experimental data, the dashed lines represent the fitting lines of the main melting peaks, and the dotted lines represent the fitting lines of the shoulder melting peaks. Curve (a) is for 10 parts, (b) is for 15 parts, (c) is for 20 parts, (d) is for 25 parts, and (e) is for 30 parts of thermoplastic starch.

**Figure 4 polymers-15-01579-f004:**
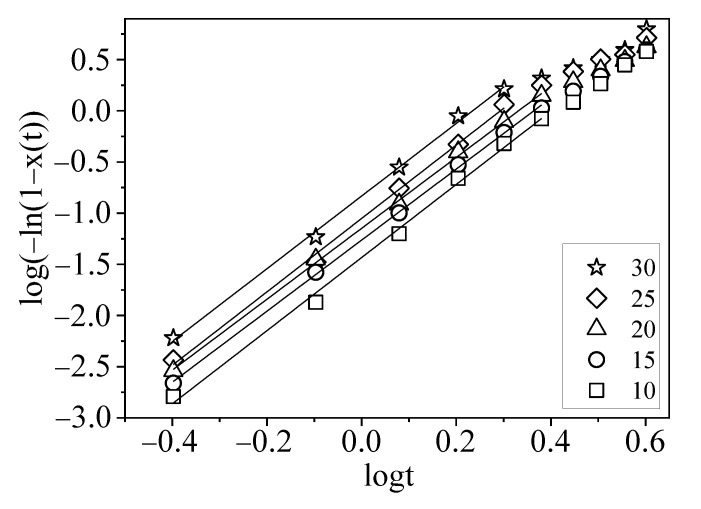
Plots of $log\left[-ln\left(1-X\left(t\right)\right)\right]$ versus logt for the nonisothermal crystallization. The symbols represent the experimental data, and the solid lines represent the fitting lines of the primary crystallization. The symbol □ is for 10 parts, ○ is for 15 parts, △ is for 20 parts, ◇ is for 25 parts, and ☆ is for 30 parts of thermoplastic starch.

**Figure 5 polymers-15-01579-f005:**
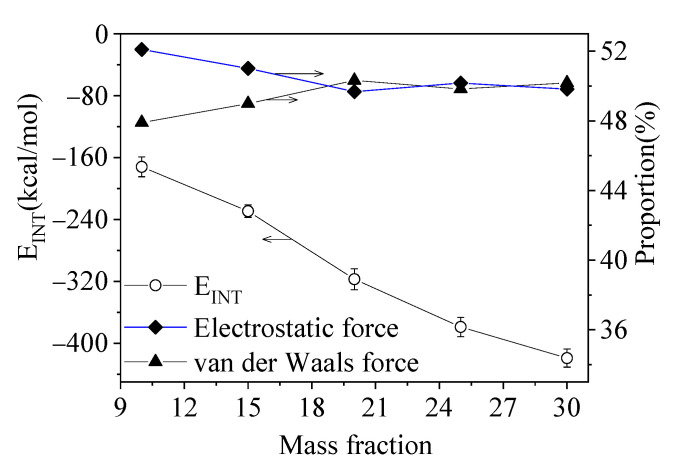
The relationship between the interfacial binding energy, the electrostatic force, and van der Waals force with the mass fraction of thermoplastic starch.

**Figure 6 polymers-15-01579-f006:**
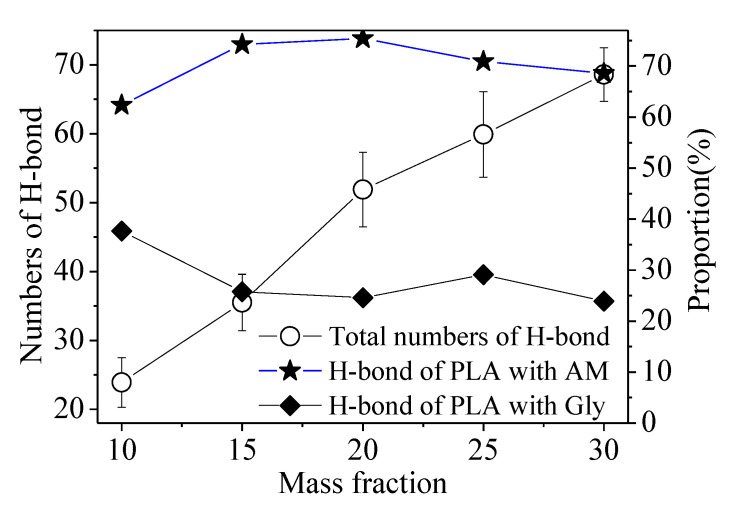
The relationship between the total number of H-bonds, the proportion of H-bonds of PLA-AM, and the proportion of H-bonds of PLA-Gly with the mass fraction of thermoplastic starch.

**Table 1 polymers-15-01579-t001:** The thermal transition temperatures of PLA/thermoplastic starch/SiO_2_ composites.

Mass Fraction of Thermoplastic Starch	Prehistoric			Elimination of Thermal History		
	T_g_/°C	T_m_/°C		T_g_/°C	T_m_/°C	
0	64.1	-	167.3	60.4	-	165.8
10	50.6	139.2	153.7	42.9	143.7	155.6
15	50.2	139.3	152.7	46.8	144.9	155.6
20	51.6	139.5	152.8	49.4	145.1	155.7
25	52.6	140.6	152.5	51.4	145.3	155.1
30	52.5	142.0	152.1	51.7	146.1	155.5

**Table 2 polymers-15-01579-t002:** The relative crystallization of PLA on its interface with thermoplastic starch and in its matrix.

Mass Fraction of Thermoplastic Starch	T_c,onset_ /°C	T_c_ /°C	Crystallinity /%	Relative Proportion of Crystallization		
				$X_{c,interf}$/%	$X_{c,m}$/%	$R^{2}$
0	104.5	134.1	13.3	0	100	---
10	86.7	92.2	28.1	5.8	94.2	0.9667
15	91.5	97.6	30.0	11.1	88.9	0.9686
20	92.2	98.9	31.2	13.8	86.2	0.9757
25	92.8	99.8	38.3	17.7	82.3	0.9812
30	93.2	102.6	42.1	19.2	80.8	0.9808

**Table 3 polymers-15-01579-t003:** The kinetics parameters of the fitting of the primary crystallization.

Mass Fraction of Thermoplastic Starch	n	logZ	$Z_{c}$	$R^{2}$
10	3.57	−1.43	0.72	0.9957
15	3.47	−1.26	0.75	0.9994
20	3.46	−1.15	0.77	0.9988
25	3.59	−1.05	0.79	0.9968
30	3.55	−0.83	0.83	0.9974

## Data Availability

The data presented in this study are available on request from the corresponding author.

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
