# Peer review of "The Role of the Interface of PLA with Thermoplastic Starch in the Nonisothermal Crystallization Behavior of PLA in PLA/Thermoplastic Starch/SiO2 Composites"

_polymers, 2023, doi:10.3390/polym15061579_

Round 1

Reviewer 1 Report

The peer-reviewed article titled The role of the interface of PLA with thermoplastic starch in the nonisothermal crystallization behaviour of PLA in PLA/thermoplastic starch/SiO2 composites presents the effect of the applied systems on the change of the crystal structure.

In my opinion, the subject matter undertaken is in line with current material trends and consistent with the scope of the journal Polymers. Nonetheless, as a reviewer, I have some comments on the manuscript.

My comments:

The authors did not show the research novelty of the paper. Please supplement.

Information about the structure of PLA and the value of its MFI is missing.

Was PLA dried before processing? There is no such information in the manuscript.

What was the form of the extrudate that was obtained?

What did the sample look like for DSC studies? How was it collected?

Please complete and systematize in the methodology the composition of the polymer compositions obtained, the convention adopted makes the analysis of the results much more difficult.

The article indicates two ways of mixing/dispersing nano Sio2 with glycerol, please systematize.

Line 168 -170: “When the mass fraction of PLA and thermoplastic starch are the same, the composites exhibit similar thermal transition temperatures and crystallization temperature due to their similar interface of PLA with thermoplastic starch” On the basis of which studies these conclusions were made?

Line 173-175 “It is dispersed in PLA matrix, which prevents its crazing and its following development, and thus effectively improves the rheological properties and the mechanical strength of the composites” On the basis of which studies these conclusions were made?

No DSC curve for unmodified PLA (fig. 1).

Could the reduction in Tg be due to the higher affinity of glycerol for PLA and its plasticizing role in this polymer?

What does the term composites with thermal history mean? What is the thermal history (cooling conditions) of the tested materials Differentiating samples with and without thermal history can be interchanged as an analysis of the first heating cycle or the second heating cycle when the thermal history is not specified

The inference should be supplemented with the effect of nano-Sio2 on PLA.

Author Response

Dear reviewer:

Thank you for your very accurate and useful comments to improve our manuscript.  I have revised the manuscript according to your comments. We have highlighted in red the changes introduced in the manuscript.

Thank you and best regards.

Yours sincerely,

Deling Li

Reviewer 2 Report

Grammar/Spelling

The manuscript's writing, phrasing, and word choice need work; there are too many meaningless sentences that don't contribute anything to the discussion. Some sentences need to be improved grammatically as

1.    The abstract fails to provide a summary of the most important parts of this manuscript.

2.    The language used to describe the experimental materials and methods is all present tense, while it should be written in the past tense.

3.    line no. 35, as per present indefinite tense, word should be ‘limits’

4.    line no. 69, Nano-SiO2 is Aerosil 200 (Degussa) with a specific surface area.….. (it is not clear from start)

5.     line no. 81 and 87, “are” should be “is” as single temperature profile is mentioned.

6.    line no. 93, both sentences should have a connection.

7.     At section 2.5 Simulation part, second line of the paragraph needs rephrasing. 

8.    Line no. 148 needs improvement.

9.    Line no. 153 is not making connection and sense with the previous data as “Their crystalline structure of starch is destroyed”

10.Add references for data reported in literature at line 154 of the results section and elsewhere.

11.  In conclusion section, it should be “increases” in line no. 325.

12.  Needs improvement in writing of line no. 174

13. At line no. 308, why abbreviation AM and GLY are reported again? 

Following points need attention

Literature review is missing in introduction in the manuscript.

Exact amount of the material used in the preparation of ‘thermoplastic starch/SiO2 is not clearly reported as “An appropriate amount of corn starch is mixed into the suspension of nano-SiO2 with glycerol….

In heading 2.3, again a certain amount of PLA is mentioned which should be a clear and exact numerical value.

Caption of figure 1 should be precise for a, b, c, d curves which should be written in brackets as (a), (b)

References should be provided for the decrease of PLA free volume interchain during melt extrusion process. (line no. 183)

Figure 2 needs uniform font size labeling and caption improvement.

Reference should be provided for relation of mass fraction with Xc and Xc,m.

Table no. 2 needs formatting as words are splitting in column (Crystallinity).

References should be provided for the section 2.5 where data is reported for results.

In Bibliography, reference no. 3 and 20 are not according to the format.

Label and captions of the tables and specifically figure needs improving. Details of the figure contents should be reported in separate below paragraphs.

Conclusion is not reflecting and summarizing the manuscript.

Author Response

(The authors gave the same response as above.)

Reviewer 3 Report

THe paper is on the role of PLA interface on non-isothermal crystallization of PLA in PLA/thermoplastic starch composite. I found the paper interesting, and I enjoyed reading it. I have few question for author thouhg. 

1. What was the main reason for clearing the thermal history in the composite?

2. Is there a way to quantify the interface in the composite? Also, a small sample is used for DSC, and the composite is usually not homogeneous. Is there any possibility of bias in the results regarding the selection of the DSC samples?

Author Response

(The authors gave the same response as above.)

Round 2

Reviewer 1 Report

I thank the authors for their responses to my comments. I support the authors' request that the peer-reviewed manuscript in its current form be accepted for further editorial stages.

Reviewer 2 Report

Authors have made sufficient changes in the draft